# The relationship between line manager training in mental health and organisational outcomes

**Juliet Hassard**[1][☉], **Teixiera Dulal-Arthur**[2], **Jane Bourke**[3], **Maria Wishart**[4], **Stephen Roper**[4], **Vicki Belt**[4], **Stavroula Leka**[5], **Nick Pahl**[6], **Craig Bartle**[2], **Louise Thomson**[2,7], **Holly Blake**[8,9][☉] *

**1** Queen's Business School, Queen's University Belfast, Belfast, Northern Ireland, United Kingdom, **2** School of Medicine, University of Nottingham, Nottingham, United Kingdom, **3** Cork University Business School, University College Cork, Cork, Ireland, **4** Warwick Business School, Warwick University, Coventry, United Kingdom, **5** Centre for Organisational Health & Well-being, Lancaster University, Lancaster, United Kingdom, **6** The Society of Occupational Medicine, London, United Kingdom, **7** Institute of Mental Health, University of Nottingham, Nottingham, United Kingdom, **8** School of Health Sciences, University of Nottingham, Nottingham, United Kingdom, **9** NIHR Nottingham Biomedical Research Centre, Nottingham, United Kingdom

☉ These authors contributed equally to this work.
* holly.blake@nottingham.ac.uk

**Data Availability Statement:** The dataset used in this study is available from the Nottingham Research Data Management Repository (http://doi.org/10.17639/nott.7419).

## Abstract

### Background

Line manager (LM) training in mental health is gaining recognition as an effective method for improving the mental health and wellbeing of workers. However, research predominantly focuses on the impacts of training at the employee-level, often neglecting the broader organisational-level outcomes. Most studies derive insights from LMs using self-reported data, with very few studies examining impacts on organisational-level outcomes.

### Aim

To explore the relationship between LM training in mental health and organisational-level outcomes using company-level data from a diverse range of organisations.

### Methods

This study is a secondary analysis of anonymised panel survey data from firms in England, with data derived from computer-assisted telephone surveys over four waves (2020, 1899 firms; 2021, 1551; 2022, 1904; and 2023, 1902). The analysis merged the four datasets to control for temporal variations. Probit regression was conducted including controls for age of organisation, sector, size, and wave to isolate specific relationships of interest.

### Results

We found that LM training in mental health is significantly associated with several organisational-level outcomes, including: improved staff recruitment (β = .317, p < .001) and retention (β = .453, p < .001), customer service (β = .453, p < .001), business performance (β =

**Funding:** The data used here were originally collected as part of an Economic and Social Research Council funded project 'Workplace mental-health and well-being practices, outcomes and productivity' (Grant number: ES/W010216/1). This secondary analysis project 'Mental health at work: a longitudinal exploration of line manager training provisions and impacts on productivity, individual and organizational outcomes' was supported by the Economic and Social Research Council [The Productivity Institute: grant number: ES/V002740/1]. There was no additional external funding received for this study.

**Competing interests:** The authors have declared that no competing interests exist.

.349, p < .001), and lower long-term sickness absence due to mental ill-health (β = -.132, p < .05).

## Conclusion

This is the first study to explore the organisational-level outcomes of LM training in mental health in a large sample of organisations of different types, sizes, and sectors. Training LM in mental health is directly related to diverse aspects of an organisations' functioning and, therefore, has strategic business value for organisations. This knowledge has international relevance for policy and practice in workforce health and business performance.

## Introduction

In the United Kingdom (UK), one in six workers experience mental health challenges, with 12.7% of all sickness absence days attributed to mental ill-health [1]. The estimated annual cost of poor employee mental health to British employers is £56 billion annually [2]. The importance of employers addressing mental health in the workplace is emphasised in both national (e.g., 'Mental Wellbeing at Work', National Institute for Health and Care Guidance [3]) and international (e.g., International Organization for Standardization 'ISO 45003' standards on psychological health and safety at work [4]) policies through the implementation of workplace mental health and wellbeing (MH&WB) practices by organisations.

There are diverse MH&WB practices that employers may utilise to promote mental health at work and prevent work-related stress, each with a different target of change. The IGLO model [5] identifies targeted areas of (behavioural and organisational) change necessary to promote mental health at work:

1. Remedial and resiliency focused strategies targeting **I**ndividual behaviour change and health management (e.g., stress management, mindfulness).

2. **G**roup-level strategies targeting improved social support, work group climate, and increased knowledge and understanding of mental health (e.g., team building exercises).

3. Improving managers' and **L**eaders' knowledge, skills, and abilities to promote mental health among those they manage (e.g., line manager (LM) training).

4. Improving **O**rganisations through human-centric working conditions and enhanced job quality, underpinned by a psychosocial risk assessment (e.g., job design, job crafting, flexible work arrangements).

The impact and value of three of these targeted areas of change (individual, group and organisational) has been demonstrated in a burgeoning literature for both individual- (e.g., improved health and work motivation; [6–8]) and organisational-level outcomes (e.g., reduced sickness absence and decreased turnover; [9,10]).

The Job Demand Resource Model (JDR; Fig 1) [11,12]) provides a theoretical framework to understand the conceptual link between working conditions, employee mental health, and work performance and productivity outcomes. The JDR model postulates that work characteristics (categorised as either job demands or job resources) influence workers' psychological well-being and work engagement. *Job demands* are those factors that require emotional or cognitive effort, which can result in psychological or physical harm. Conversely, *job resources* refer

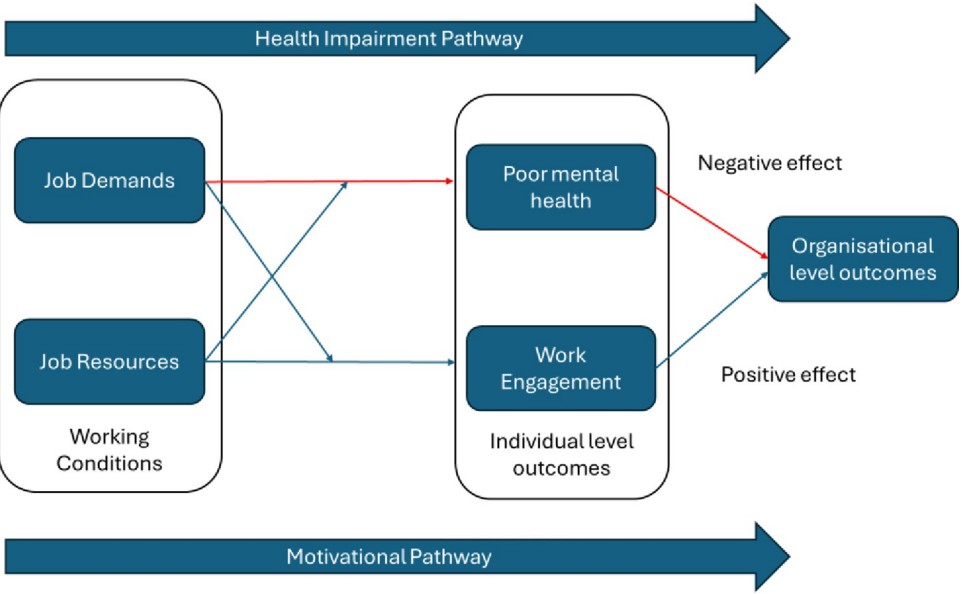

**Fig 1. Job Demand Resource Model.**

to those physical, social, or organisational aspects of the job that may: reduce job demands and their associated physiological and psychological costs; be functional in achieving work goals; and stimulate personal growth, learning, and development [11,12]. This model postulates two pathways that help to understand the link between employees' working conditions, mental health and well-being, and organisational-level outcomes (e.g., customer service, productivity, absenteeism).

The *health impairment pathway* postulates that high job demands negatively impact employee mental health and, by extension, results in poor organisational-level outcomes. Conversely, *the motivational pathway* hypothesises that high levels of job resources improve employee motivation and engagement, and by extension, results in better organisational-level outcomes. Meta-analytic evidence using longitudinal studies finds strong evidence of the link between job demands and resources to individual-level health and motivational outcomes [13]. However, they emphasise the paucity of data examining the link between employee well-being and organisational-level outcomes, highlighting this as a key and pervasive gap in knowledge.

MH&WB practises aim to target each of these pathways in slightly different ways. From enhancing employees' health management skills, to improving and supporting their engagement and performance at work (individual-focused) to the (re)design of work environments, to minimising job demands and enhancing opportunities for job resources. Therefore, understanding and testing the postulated link between MH&WB practises and organisational level outcomes addresses an important gap in knowledge. This is particularly true of interventions targeting leaders and managers.

In 2010, Kelloway and Barling [14] highlighted the need for, and value of, manager-focused interventions to support workplace mental health promotion. More recently, manager training in mental health was specifically identified as a strong recommendation for interventions by the World Health Organization (WHO) in their guidelines on mental health at work [15]. This approach to mental health training is unique and different from Mental Health First Aid (MHFA) training [16], which has become increasingly popular and researched in recent years.

MHFA training aims to upskill nominated employees from across the organisation to provide support for a work colleague who is developing a mental health problem, experiencing a worsening of a mental health problem, or is in a mental health crisis. While increasingly popular in practice, recent evidence highlights a lack of high-quality evaluative data investigating its impact on employees or the organisation [17]. MHFA training is focused on remedial care and support, in contrast to prevention focused efforts to improve employees' working conditions and management practices through the development and upskilling of LMs.

Therefore, the provision of LM training in mental health has emerged as a viable approach to improving the mental health of workers. With the rise in the prevalence of mental ill-health in the UK during the coronavirus (COVID-19) pandemic, there has since been a notable increase in the number of organisations offering LM training for mental health (2020: 50%; 2023: 59% [18]). LM training in mental health is a systematic approach to equipping LMs with the knowledge, skills, and attitudes needed to support the mental health of their team members and individuals they line manage [19]. This training may include a focus on the LMs' *own* mental health and wellbeing, as well as that of the individuals they manage [20].

Despite its increasing popularity in practice over recent years, there remains a limited (but growing) literature that has sought to investigate the impact of LM training interventions on employees directly, and indirectly on their organisations. Employee-level benefits of LM training in mental health have previously been reported [20–22]. Such benefits include increased behavioural competencies to support those in their team with mental health challenges [20], behavioural intentions to promote mental health at work [22], and increased confidence to support the mental health of those they manage [21].

Comparatively fewer studies (e.g., [21,23]) have examined the impact and influence of LM training on organisational-level outcomes (e.g., changes to productivity, turnover rates, and absenteeism). To our knowledge, no studies to date have explored these relationships across a diverse range of organisations by size and sector and drawing on company-level (rather than self-reported employee-level) data. Understanding the impact of LM training at both employee- and organisational-levels is vital to appreciating its empirical and practical value in promoting mental health at work. In particular, understanding the organisational-level impact and benefit of LM training informs the business case for workplace mental health promotion. Understanding the economic arguments and benefits of workplace mental health promotion is an important motivator for employers to implement such strategies [24,25].

The aim of our study was, therefore, to investigate the association between the provision of LM training for mental health in organisations and organisational-level outcomes. This was achieved by utilising company-level data derived from an existing longitudinal survey of employers in England examining their MH&WB practices. The research questions (RQs) and hypotheses (Hs) are outlined below:

RQ1: Do organisations that offer LM training in MH have more, or less, sickness absence due to mental ill-health compared with organisations that do not offer this?

H1: Organisations that offer LM training in MH will have less sickness absence due to mental ill-health compared with organisations that do not offer this.

RQ2: How do organisations offering LM training in MH compare with those that do not, on organisational-level outcomes (e.g., recruitment, retention, customer services, business performance).

H2: Offering LM training in MH will be associated with improved organisational-level outcomes (e.g., recruitment, retention, customer services, business performance).

## Materials and methods

This study is based on a secondary analysis of anonymised panel survey data from firms in England. Reporting was guided by the Strengthening the Reporting of Observational Studies in Epidemiology (STROBE) Statement [26]) (Supplementary file 1) and the Consensus-Based Checklist for Reporting of Survey Studies (CROSS) [27] (Supplementary file 2). The data were derived from UK Computer Assisted Telephone Interview (CATI) surveys (a commonly used approach for reaching business personnel), collected over four time periods. Telephone surveys were used to reduce non-response bias commonly associated with mailed surveys [28].

Interviews were conducted by call centre operatives from a UK-based independent market research company. All telephone interviewers were independently evaluated throughout the fieldwork period, using a scorecard covering all aspects of their interview. The evaluation was based either on supervisory live listening or via audio recordings of the interview. CATI processes are evaluated annually by ISO20252 standard (for market, opinion, and social research) auditors and a minimum of 10% of interviews from each interviewer are evaluated and documented. In this study, approximately 12%-14% of interviews were subject to live listening quality control (QC), with around 5–10% of interviews undergoing full QC (listening to recordings and checking data once the survey is complete). All interviewers were trained in research methods and undertook a half-day training session prior to the study starting, involving role play and survey piloting to identify and resolve ambiguities. To minimise human error in data entry, data checking was undertaken, and outliers were identified and checked.

The data used in this study were collected in four waves as part of a broader prospective study which is funded by the UK Economic and Social Research Council and is ongoing. Wave 1: 6 January to 20 March 2020 (1899 firms), Wave 2: 28 January to 15 April 2021 (1551 firms), Wave 3: 27 January to 20 May 2022 (1904 firms) and Wave 4: 16 January to 5 May 2023 (1902 firms) all including data from non-government funded organisations with 10 or more employees. Organisations were additionally screened to ensure: (a) they were not a local or central government financed body; (b) they had been trading for three or more years. Branches and subsidiaries of larger businesses were included in the survey. Organisations with 10–19 employees were intentionally under-sampled as they accounted for most of the population universe. Larger organisations were over-sampled to ensure they were adequately represented, to reduce sampling bias and to allow more robust sub-analysis. The intention was to obtain as broad a response as possible during the data collection period, and so the final sample was the number of participants that responded between the survey opening and closing dates for each wave. Within each organisation, the most senior person with responsibility for the health and wellbeing of workers was approached and invited to participate as a representative of that organisation.

Organisations participating in Wave 1 were followed up in subsequent waves by the call centre operatives until an appointment was made or the organisation refused. However, as the study utilised unbalanced panel data rather than longitudinal data, new organisations were recruited at each wave to increase the overall sample size. In total, 118 organisations participated in the study across all four waves. The research was conducted in line with the Declaration of Helsinki. Ethical approach for this analysis was granted in August 2023 by the institutional Research Ethics Committee (Ref: HSSREC-144 21–22). Participants in the surveys provided informed oral consent which was documented by the telephone operatives. Although this is not clinical research, the senior author (HB) is trained in Good Clinical Practice (GCP). All researchers were trained in research ethics and research methods. The surveys were anonymised, and all datasets were stored on password protected files and only shared using a password protected OneDrive shared folder. The datasets were accessed on 23 August 2023.

**Table 1. Operationalisation of study variables.**

| Construct | Item Description | Measurement Level | Sample Size (n) |
|---|---|---|---|
| **Sickness absence due to mental ill-health** | | | |
| Presence of sickness absence due to mental ill-health | In the last 12 months, have any staff been off sick, for any length of time, due to mental health problems, including illnesses such as bipolar disorder, depression, anxiety, or stress? | Binary | 3385 |
| Proportion of sickness absence due to mental ill-health | What proportion of sickness absence over the last 12 months was accounted for by mental health problems? | Continuous | 1116 |
| Repeated sickness absence due to mental ill-health | In the last 12 months have you had any instances where staff took repeated sickness absence because of mental health problems? | Binary | 3566 |
| Proportion of long-term sickness absence due to mental ill-health | What proportion of sickness absence due to mental health problems over the last 12 months has been long term (a single absence lasting 4 weeks or more)? | Continuous | 3566 |
| **Organisational-level outcomes** | | | |
| Staff recruitment | Helped with staff recruitment | Binary | 3182 |
| Customer service | Improved customer service | Binary | 3178 |
| Staff retention | Improved staff retention/ reduced staff turnover | Binary | 3189 |
| Business performance | Improved business performance | Binary | 3185 |

Table 1 provides a summary of the operationalisation of our study variables. Our predictor variable was 'LM training in mental health'. Quantified as a single, dichotomous variable (coded: no = 0, yes = 1). We tested four organisational-level outcome variables examining staff recruitment, customer service, retention, and business performance. All four variables were measured using a single-item question measured categorically (yes/no).

Four items quantified sickness absence related to mental ill-health trends, within the last 12 months, for the organisation. These four items allowed us to examine: (1) the presence (or absence) of sickness absence due to mental ill-health in the organisation; (2) the proportion of all sickness absence cases accounted for by mental ill-health; (3) the presence (or absence) of repeated sickness absence cases; and (4) the proportion of sickness absence for mental ill-health that was long-term (>4 weeks). Two items were measured dichotomously (Yes/ No), and two by a reported percentage.

For analysis purposes, the two sickness absence items measured continuously were dichotomised into 'high' and 'low' classifications. For the proportion of sickness absence (high/low), we utilised the sample mean for each annual survey wave to determine the numeric thresholds delineating high (> sample mean) and low (< = sample mean) classifications: 2020, 17%; 2021, 20%; 2022, 19%; and 2023, 22%. The same approach was utilised to specify 'high' and 'low' categorisations for our study variable examining the proportion of recurrent, long-term cases of sickness absence due to employee mental ill health within the organisation: 2020, 17%; 2021, 49%; 2022, 45%; and 2023, 40%.

We conducted probit regression analyses using SPSS Version 28 (Armonk, NY: IBM Corp) to determine the probability of specific outcomes occurring based on the presence or absence of LM training in mental health, allowing a deeper understanding of how LM training for mental health predicts various organisational-level outcomes. This analysis was deemed most appropriate due to its capacity to model binary outcomes, specifically yes/no responses, by quantifying probabilities rather than odds ratios [29]. Given the data structure of the study, probit regression was selected in preference to other viable alternatives such as logistic regression.

The four waves of data were merged for the analyses. This approach yielded several benefits such as providing increased statistical power through a larger sample size [30], allowing us to assess the stability and consistency of the relationships over time [31], and enhancing the overall generalisability of the results beyond the specific year in which data were collected [32]. As pooled panel data were used, we did not employ specific strategies to address missing data in the analysis as any missing data points were inherently handled through the nature of the dataset.

The regression analysis controlled for age of the organisation [0–10 years, 11–20 years, more than 20 years], sector [Production, Construction, Wholesale/Retail, Hospitality, Business Services and Other Services], size of the organisation [1–49 employees; 50–249 employees and 250+ employees] and survey wave [2020, 2021, 2022 and 2023]. This recognises that the age of the organisation, the specific sector in which an organisation operates, and its size can impact its performance, culture, decision-making processes, management practices and overall behaviour [33,34]. By controlling for these variables, we account for these differences ensuring a more accurate understanding of the relationship between the predictor and outcomes. The number and typology of organisations offering LM training in mental health (based on analysis of the same dataset) is detailed elsewhere [18].

## Results

Table 2 provides an overview of our sample across several key demographics. After merging the four waves of data we had a final sample of n = 7139. The highest proportion of participating organisations came from the business services sector, operated for more than 20 years, and fell within the micro-to-small size category. The response rates were 17% (2020) and 15% (2021–2023). Response rate was calculated as the percentage of people who completed and answered the survey out of the total number of people invited to take part.

### RQ1: Do organisations that offer LM training in MH have more, or less, sickness absence due to mental ill-health compared with organisations that do not offer this?

To investigate RQ1, we used probit analysis to test whether organisations that offered LM training in mental health have lower reported levels of sickness absence (including, the

**Table 2. Characteristics of participating organisations.**

| Characteristics | 2020 (n = 1899) | 2021 (n = 1551) | 2022 (n = 1904) | 2023 (n = 1902) |
|---|---|---|---|---|
| **Sector** | | | | |
| Production | 364 (19.2%) | 362 (23.3%) | 411 (21.6%) | 414 (21.8%) |
| Construction | 139 (7.3%) | 111 (7.2%) | 145 (7.6%) | 136 (7.2%) |
| Wholesale, retail | 402 (21.2%) | 331 (21.3%) | 364 (19.1%) | 363 (19.1%) |
| Hospitality | 204 (10.7%) | 109 (7.0%) | 187 (9.8%) | 210 (11.0%) |
| Business services | 468 (24.6%) | 350 (22.6%) | 431 (22.6%) | 419 (22.0%) |
| Other | 322 (17.0%) | 288 (18.6%) | 366 (19.2%) | 360 (18.9%) |
| **Length of Operation** | | | | |
| 0–10 years | 301 (16.0%) | 236 (15.3%) | 263 (13.9%) | 250 (13.2%) |
| 11–20 years | 508 (27.0%) | 378 (24.5%) | 510 (27.0%) | 541 (28.7%) |
| 20 + years | 1072 (57.0%) | 929 (60.2%) | 1115 (59.1%) | 1097 (58.1%) |
| **Size of Organisation** | | | | |
| Micro-small (< = 50 employees) | 1445 (76.1%) | 1225 (79.0%) | 1537 (80.7%) | 1579 (83.0%) |
| Medium (51–250) | 367 (19.3%) | 286 (18.4%) | 310 (16.3%) | 286 (15.0%) |
| Large (>250) | 87 (4.6%) | 40 (2.6%) | 57 (3.0%) | 37 (1.9%) |

**Table 3. Probit regression testing the associations between LM training for MH (y/n) and organisational-level sickness absence trends due to mental ill-health.**

| Outcomes | Results |
|---|---|
| Presence of sickness absence due to mental ill-health (n = 3385) | β .160 (.0471) |
| | LR chi$^2$ 286.805*** |
| | Log likelihood −493.883 |
| Proportion sickness absence due to mental ill-health (n = 1116) | β -.077 (.0525) |
| | LR chi$^2$ 41.654*** |
| | Log likelihood −414.569 |
| Repeated sickness absence due to mental ill-health (n = 3566) | β .027 (.0803) |
| | LR chi$^2$ 49.490*** |
| | Log likelihood -339.804 |
| Proportion of long-term sickness absence due to mental ill-health (n = 3566) | β -.132* (.0577) |
| | LR chi$^2$ 388.557*** |
| | Log likelihood −390.677 |

Note 1: Analysis controlled for wave, sector, size, and age of organisation.

Note 2: Standard error placed in brackets.

Note 3: LR chi$^2$ = Likelihood ratio chi-square.

presence (absence of) sickness absence cases and repeated cases, and high or low levels of sickness absence cases for mental health and long-standing cases (> 4 weeks; see Table 3). We observed only one significant association among the four areas of sickness absence trends tested. We found that offering LM training in mental health significantly predicted below-average levels of long-term sickness absence due to mental ill-health. Significant associations were not observed across the three other examined outcome variables for sickness absence.

The bar chart in Fig 2 below displays the regression coefficients which demonstrate the strength and directions of the associations between LM training for MH and the various outcomes related to sickness absence due to mental ill-health, including the presence, proportion, repeated occurrences, and long-term proportions.

### RQ2: How do organisations offering LM training in MH compare with those that do not, on organisational-level outcomes (e.g., recruitment, retention, customer services, business performance)

To explore RQ2, we used probit analysis to test whether organisations that offered LM training in MH (as compared to those that did not) reported better efforts to recruit new staff, and improved customer service, staff retention and business performance (see Table 4). We found organisations offering LM training were more likely to report a range of positive outcomes including help with staff recruitment, improved customer service, improved staff retention and improved business performance.

The bar chart in Fig 3 below shows the regression coefficients (β) for the associations between LM training for mental health (MH) and improved organisational-level outcomes, including staff recruitment, customer service, staff retention, and business performance. Each bar represents the β coefficient value, indicating the strength and direction of the associations.

Table 5 below summarises these results and highlights how it compares to previous research and the unique contributions this study brings to the existing literature.

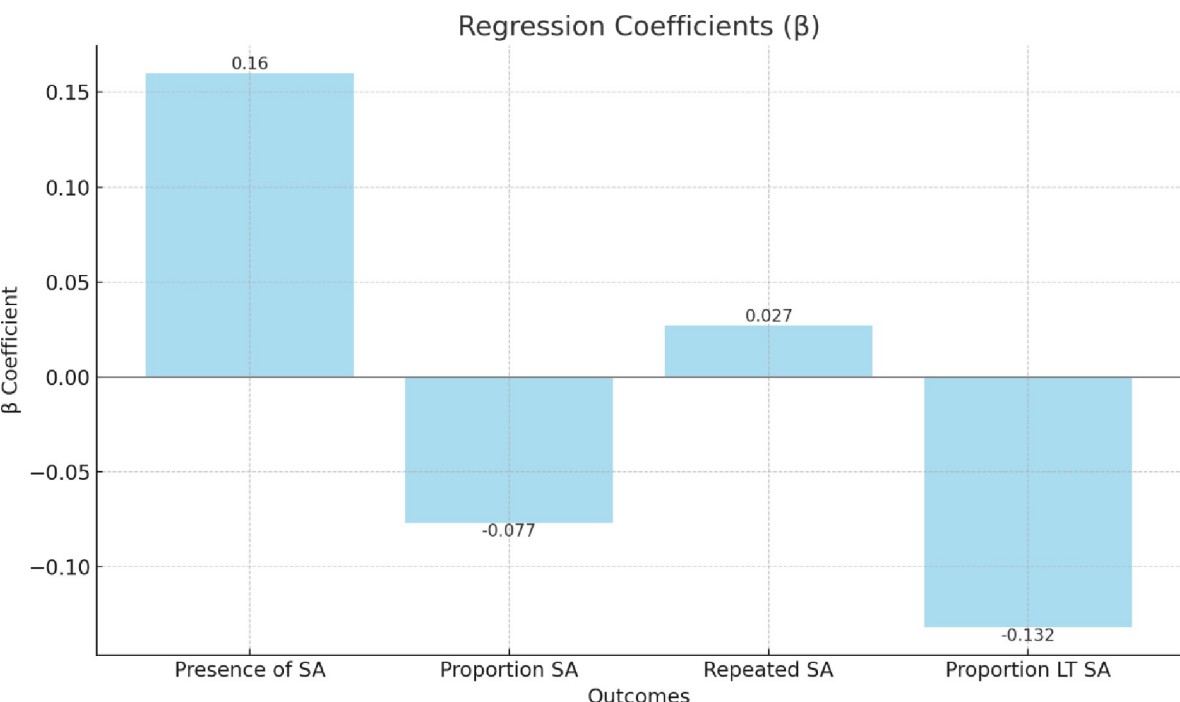

**Fig 2. Probit regression testing the associations between LM training for MH (y/n) and organisational-level sickness absence trends due to mental ill-health.** Note: SA = Sickness absence due to mental ill-health; LT SA = Long-term sickness absence due to mental ill-health.

**Table 4. Probit regression analysis testing the associations between LM training for mental health (y/n) and improved organisational-level outcomes.**

| Dependent Variables | Results |
|---|---|
| Staff recruitment (n = 3182) | β .317*** (.0467) |
| | LR chi² 118.377*** |
| | Log likelihood –484.131 |
| Customer service (n = 3178) | β .453*** (.0485) |
| | LR chi² 184.147*** |
| | Log likelihood –451.901 |
| Staff retention (n = 3189) | β .379*** (.0479) |
| | LR chi² 95.828 |
| | Log likelihood –476.803 |
| Business performance (n = 3185) | β .359*** (.0496) |
| | LR chi² 105.963 |
| | Log likelihood –446.303 |

Note 1: Analysis controlled for wave, sector, size and age of organisation.

Note 2: Standard error placed in brackets.

Note 3: LR chi² = Likelihood ratio chi-square.

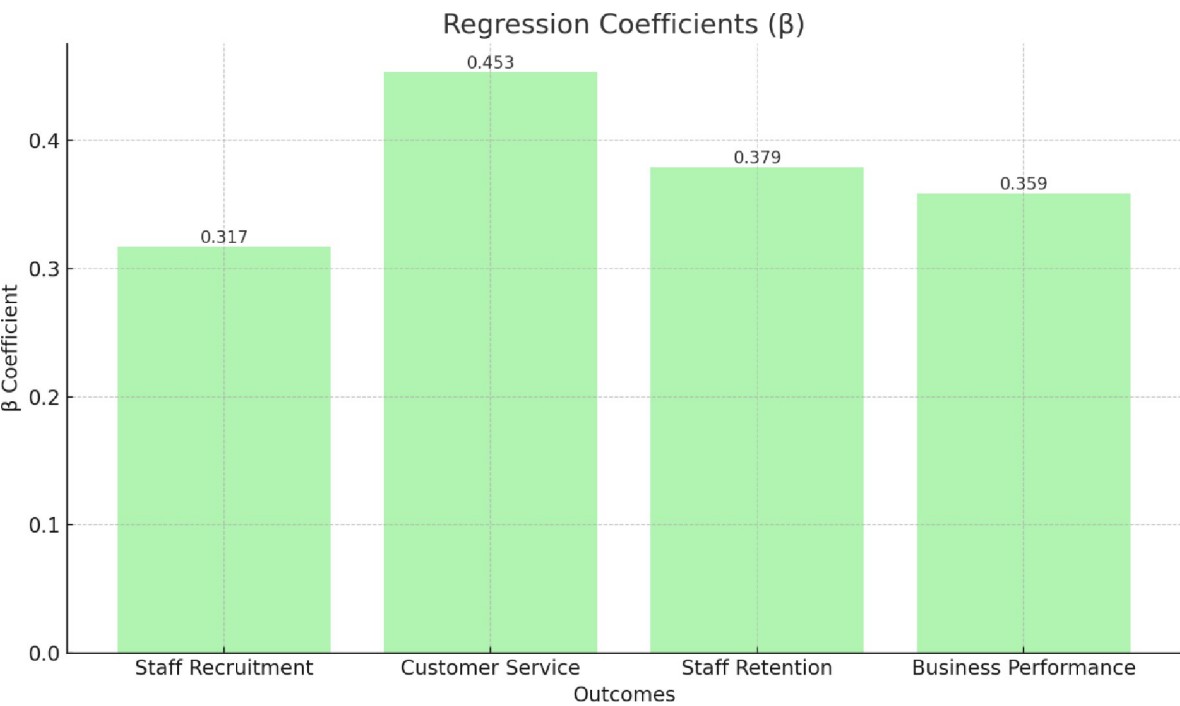

**Fig 3. Probit regression analysis testing the associations between LM training for MH (y/n) and improved organisational-level outcomes.**

## Discussion

To our knowledge, this study was the first to investigate the provision of LM training for mental health in organisations and its association across organisational-level outcomes including sickness absence, staff recruitment and retention, customer service, and business performance. We found that organisations that train their line managers in mental health have better outcomes across all these areas, which has national and international relevance to research, policy, and practice in workplace health.

A key empirical strength of this study is that we investigated these associations within a diverse sample of British organisations (e.g., company size, type, and sector), using company-level data. This data set allowed us to, uniquely, explore the often discussed and postulated

**Table 5. Summary of key findings, comparison to previous research and research contributions.**

| Key Findings | Comparison to Previous Research | Research Contributions |
|---|---|---|
| LM training in mental health was significantly associated with below-average levels of long-term sickness absence due to mental ill-health. | Evidence shows that LM training in mental health may improve managers' self-reported capabilities to support employees with mental ill-health following return to work [21,22]. | Provides an empirical example of the Chain of Impact Model by showing how the provision of LM training for mental health may translate organisation implemented interventions into improved economic outcomes for the organisation. |
| LM training in mental health was not associated with short-term sickness absence due to mental ill-health or repeated sickness absence due to mental ill-health. | One single site study showed that LM training in mental health reduced work-related sickness absence [21]. | Our findings highlight the complexity of LM training's relationship with various kinds of sickness absences and distinguishes between its relationship with other categories of mental health related absences. |
| Organisations offering LM training were more likely to report improvements in staff recruitment, customer service, staff retention and business performance. | Research shows that LM training in mental health can lead to improvements in employee-level outcomes [20]. | Extends understanding of the benefits of LM training beyond employee well-being to include organisational benefits, thus broadening our understanding of the broader impact of LM training. |

relationship between mental health and well-being practises in the workplace and wider organisational performance indicators (e.g., [2,35–37]), some of which have been rarely examined in a systematic way within the wider workplace health management literature (e.g., customer service and business performance). As such, while the study was conducted with a sample from England, the study makes a significant contribution to the literature in this field with international relevance. Examining organisational-level outcomes is imperative as it helps to build our understanding of the business case for workplace mental health promotion and, in turn, its articulation to employers regarding its strategic business value. Therefore, our study findings make an important contribution to addressing this key gap in knowledge.

The study objective was to explore the relationship between LM training for mental health and organisational-level outcomes using company-level data from a diverse range of organisations. We observed that the provision of LM training for mental health was, on average, associated with improvement across two key organisational performance dimensions, broadly related to workforce activity (defined by dimensions of attendance, effort, quality, and innovation) and organisational outputs (defined by dimensions of productivity, business, and customer satisfaction; [38]). These two key organisational performance dimensions can be conceptually understood to drive the economic outcomes for organisations (such as profit and shareholder value), through a *Chain of Impact Model* (Fig 4). The Chain of Impact Model therefore provides a useful framework in which to interpret our findings and conceptualise how the provision of LM training for mental health may translate organisational-level outcomes into improved economic outcomes for the organisation.

We observed that, on average, the provision of LM training within organisations was associated with improved *workforce activity*, evidenced across three key indicators: below average

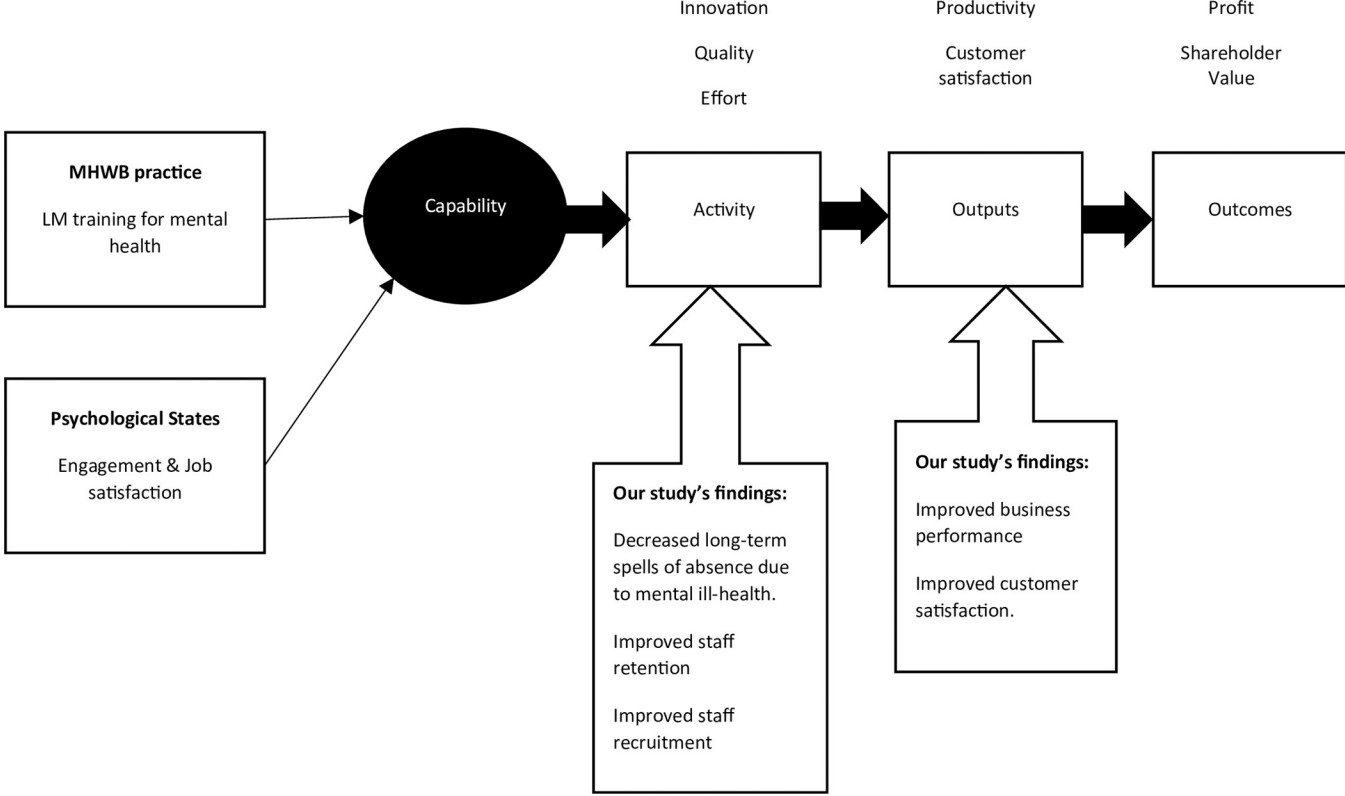

**Fig 4. Adapted Chain of Impact Model [38] based on our study focus and findings.**

long-term sickness absence spells due to mental ill-health, improved staff retention, and enhanced staff recruitment activities and efforts by the organisation. Our findings align with recent economic estimates observing the costs associated with increased staff turnover and recruitment initiatives due to poor employee mental health at work (e.g., [2]). Uniquely, our study demonstrates that the provision of LM training for mental health is associated with improved organisational metrics regarding retention and recruitment initiatives. Collectively this evidence-base highlights not only the cost of inaction in the wellbeing space, but, importantly, the potential economic benefit of investing in MH&WB practices at work.

In relation to our four indicators of sickness absence, we observe a complex and nuanced picture. We found that LM training in mental health was associated with a below average number of long-term sickness absence cases within the organisation. There is growing evidence of the impact of LM training in mental health on mangers' self-reported capabilities and confidence in supporting employees with mental ill-health during and following their return to work [21,22,39,40]. These managerial competencies are increasingly understood as a key success factor in facilitating effective return to work processes [41]. However, we did not find a significant association in relation to the three other indicators of sickness absence: the *presence* (or not) of staff off sick due to mental health problems (e.g., bipolar disorder, depression, anxiety, or stress) and *repeated cases*, or *proportion* of sickness absence accounted for by mental ill-health in general.

Reduction in these aspects of sickness absence may be primarily driven by prevention-orientated (rather than remedial-focused) MH&WB practices. Blake and colleagues [42] noted that there are few LM training initiatives that include a prevention-orientated focus (e.g., preventing stress through job quality and workload management), with most interventions targeting resiliency- or remedial-focused activities (e.g., self-care, improved knowledge surrounding mental health at work). Accounting for the nature, content and focus of LM training (rather than just its presence or not) may provide a more nuanced understanding of its association with organisational-level sickness absence indicators. We speculate that those training initiatives with a strong prevention-focus (e.g., [43]) may demonstrate a significant association with these remaining dimensions of sickness absence, as compared to those without although this is yet to be demonstrated. Testing these inferences is imperative to gathering a deeper understanding of the association between LM training and organisations' sickness absence indicators, globally.

Tamkin's Chain of Impact Model [38] theorises that these improved workforce activities should drive and influence increased organisational output metrics. While we did not test this causal pathway, we did observe that the provision of LM training for mental health was, on average, associated with improved business performance and overall customer satisfaction–both important and strategic organisational performance outcomes and precursors to improved economic outcomes. Several studies [44,45] and academic commentaries [36,37] have discussed this link, but have not tested it robustly across organisations. This is partly explained by, up until recently [46], the dearth of robust company-level data capturing both mental health and well-being practices within the organisation, as well as key productivity and performance metrics. Therefore, our early findings highlighting the association between LM training for mental health at work and strategic organisational performance outcomes are conceptually important as they provide foundational evidence for what has long been speculated regarding the costs of poor mental health at work to employers.

## Strengths, limitations and implications for future research

Workforce mental health is always an important topic, but an organisational focus on mental health was particularly pertinent during the COVID-19 pandemic when population mental-ill

health was rising, associated with myriad factors (e.g., viral transmission, economic uncertainty, social isolation). Our positive findings are therefore notable in the context of this global crisis. A significant strength of the study is its large sample size and diversity of organisations included. We are therefore able to report findings across sectors and sizes of organisations, including sectors in which mental ill-health is prevalent, but research evidence relating to workplace mental health interventions is limited (e.g., construction: [47]). Another strength of the study is that around 80% of participating organisations had fewer than 50 employees; our study therefore includes many SMEs that are under-researched and often have fewer resources to be active in mental health promotion.

Our findings provide a robust base from which to infer the potential benefits of LM training in mental health at the organisational-level and as such, are novel, and have high potential for influencing policy and practice internationally. There are several limitations within this study. First, to collect this data enterprise representatives were used to quantify workplace health and well-being practices, as well as organisational-level outcomes—including organisational performance indicators. While this is a common feature of enterprise surveys (e.g., European Survey of Enterprises on New and Emerging Risks; [48,49]), we acknowledge there may be a degree of subjective bias within this self-reporting methodology. Ideally, these associations would also be tested and triangulated with objective, rather than exclusively subjective outcomes. We view this as an important future direction for research in this field. Second, the data available within our data set allows us to test the presence (and absence) of LM training for mental health at work in relation to organisational-level outcomes. However, we anticipate that the nature of this training (in terms of its content, focus, and duration) and its perceived quality as reported by the recipients are also important explanatory variables in relation to quantifying its impact and value to the organisation. This to an important avenue of future research. Third, the use of pooled panel data hinders the capacity to capture the long-term impacts of LM training on organisational outcomes. Fourth, most research on LM training in mental health generally explores outcomes for the manager (e.g., awareness, knowledge, confidence, skills), and our study provides novel data by focusing on outcomes at an organisational level (e.g., indices of business performance). However, few studies have explored the way in which LM training is implemented in practice, and whether (or not) this impacts on mental health at the employee-level. There is currently inconclusive evidence that leadership training impacts on employee outcomes, due to contradictory results [50]. While employee outcomes were not the focus of our study, future research may seek to enhance the evidence-base relating to employee outcomes. Lastly, the low response rate observed across the four time points and the presence of missing data may indicate response bias and limit the generalisability of the findings.

## Conclusion

Training line managers in mental health is associated with better organisational-level outcomes including long-term sickness absence, staff recruitment and retention, customer service, and business performance. This study is the first to provide confirmation of the strategic value of providing LM training in mental health for organisational-level outcomes, and as such provides novel evidence with the potential to influence policy and practice internationally. Our findings strengthen the business case for organisational investment in workplace mental health and wellbeing and have relevance for diverse stakeholders including business owners and managers, professional bodies, charities, and policymakers. This study provides evidence that taking a proactive approach to workforce wellbeing has the potential to improve a diverse range of factors associated with business productivity and performance. Our primary

recommendation from this study is a clear call to action for organisations to establish work-place mental health policy that outlines the role of LMs in preventing and supporting mental health at work and invest in (or provide access to) mental health training for their LMs. This is advocated in the WHO guidelines on mental health at work [15] but not yet consistently implemented across organisation types and sectors. New, evidence-based training in mental health for LMs has been developed [42] and is currently being tested in a cluster randomised trial [23] to explore acceptability and outcomes for managers and employees.

## Supporting information

**S1 Table. STROBE statement: Checklist of items that should be included in reports of observational studies.**
(DOCX)

**S2 Table. Checklist for Reporting of Survey Studies (CROSS).**
(DOCX)

## Author Contributions

**Conceptualization:** Juliet Hassard, Holly Blake.

**Data curation:** Teixiera Dulal-Arthur, Maria Wishart, Stephen Roper.

**Formal analysis:** Teixiera Dulal-Arthur.

**Funding acquisition:** Juliet Hassard, Holly Blake.

**Investigation:** Juliet Hassard, Teixiera Dulal-Arthur, Jane Bourke, Holly Blake.

**Methodology:** Juliet Hassard, Teixiera Dulal-Arthur, Jane Bourke, Holly Blake.

**Project administration:** Teixiera Dulal-Arthur.

**Writing – original draft:** Juliet Hassard, Teixiera Dulal-Arthur, Holly Blake.

**Writing – review & editing:** Jane Bourke, Maria Wishart, Stephen Roper, Vicki Belt, Stavroula Leka, Nick Pahl, Craig Bartle, Louise Thomson.

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
