## [Decision Letter · Decision Letter 0]

3 May 2024

PONE-D-24-06886The Relationship between Line Manager Training in Mental Health and Organisational OutcomesPLOS ONE

Dear Dr. Blake,

Thank you for submitting your manuscript to PLOS ONE. After careful consideration, we feel that it has merit but does not fully meet PLOS ONE’s publication criteria as it currently stands. Therefore, we invite you to submit a revised version of the manuscript that addresses the points raised during the review process.

**Reviewer 1**The paper "The Relationship between Line Manager Training in Mental Health and Organisational Outcomes" examines the effects of managerial training on organizational outcomes. While the introduction adequately establishes the research, incorporating a discussion on existing mental health training practices could improve its context. The literature review could be strengthened by including a variety of viewpoints and conflicting research according to the variables. The methodology section, centered around probit regression, could benefit from a discussion on its limitations and potential biases and it could be analyse using confirming statistical multivariate analysis methods. The results section is well-articulated and comprehensive, though adding visual data representations and concentrating data in tables can enhance understanding. The conclusions effectively recap the study’s findings; however, elaborating on their practical applications could enhance their utility in real-world settings; increasing references to previous studies are needed to confirm research assumptions. Finally, the references are comprehensive, yet updating them to include more of the latest studies would ensure the research's relevance and show the top industry standards.

Suggestions for Enhancement:

Introduction:

I'd like you to add insights on prevailing mental health training practices in the industry to anchor the research's significance better. Include a clear definition of your research questions, objectives, and hypothesis, perhaps in a table.

Literature Review:

Broaden the scope to include diverse research and a more comprehensive array of theoretical perspectives to enrich the narrative. Explain each of your research variables and define a table of authors where you include the top publications for each variable and dimension of your study.

Methodology:

Include more details on the selection of probit regression and address any inherent methodological weaknesses or biases. Explain why it was the best alternative for analyzing this data and if there's any other possible alternative method to validate your assumptions.

Results:

I'd like you to introduce graphical elements like charts or graphs to summarize and clarify key findings, compare your results with previous research, and determine if this study covers any gap in theory, methodology, or research methods.

Conclusions:

Detail the practical implications of the research findings, suggesting specific training or policy changes that could be derived from the study. Describe the stakeholder who are having the best use for your fingins. Describe how you achieve your results and goals.

References:

Update and verify that all references reflect the most current research, ensuring the study's alignment with the latest developments in the field.**Reviewer 2**This is an interesting study investigating the organizational level effects of line manager trainings for mental health. The authors found a way to assess these outcomes by analysing data from an existing dataset (panel survey data from firms in England) thereby adding important information about the effects of these trainings that go beyond an assessment by the participants.

The manuscript is well written and clearly structured and adds important information to this area of research.

However, I have some comments that I would like the authors to respond to before publication:

The authors are right in pointing out (page 5 line 90) that only few studies have examined the impact and influence of LM training on organisational-level outcomes (e.g., changes to productivity, turnover rates, and absenteeism). However, also the impact of LM training on employee outcomes is often not assessed. This also applies to the “Employee-level benefits” that the authors describe have been found (page 5 line 85). It does not look like employees have been asked about these benefits. Instead, the results describe managers’ own assessment of their gained competencies, their intentions to promote mental health at work and their increased confidence to support employees with mental health problems. I understand that this often is what can be measured in studies about manager training as asking employees would often result in a much larger study. However, since the authors address this topic I think it is important to mention that we actually know surprisingly little about the effects of LM trainings for employees, as most studies assess how the participants themselves (the managers) evaluate what they gained from having been trained. Results often show increased knowledge, confidence and good intentions, however, we most of the time do not know if this leads to actual implementing this in daily practice (for example more support for employees with mental health problems) and we know even less about if this leads to better outcomes for employees.

This lack of knowledge of manager training for employee outcome is discussed in for example the two papers below but can also be found in several reviews about manager training for employee mental health:

Rugulies, R., Aust, B., Greiner, B. A., Arensman, E., Kawakami, N., LaMontagne, A. D., & Madsen, I. E. (2023). Work-related causes of mental health conditions and interventions for their improvement in workplaces. The Lancet, 402(10410), 1368-1381.

Aust, B., Møller, J. L., Nordentoft, M., Frydendall, K. B., Bengtsen, E., Jensen, A. B., ... & Jaspers, S. Ø. (2023). How effective are organizational-level interventions in improving the psychosocial work environment, health, and retention of workers? A systematic overview of systematic reviews. Scandinavian Journal of Work, Environment & Health, 49(5), 315.

My next comment is about missing data. The authors report on page 11, line 200 that the sample consisted of 7139 participants after the four waves were merged. However, the numbers of answers for the different questions are much lower. In Table 3 the number of responses for the sickness absence related outcomes are between 1116 and 3566. In Table 4 the numbers for the responses for the organisational-level outcomes are between 3178 and 3189. Does that mean that only around half of the participants answered these questions? Please explain.

In addition, the number of answers for the questions about the proportion of sickness absence due to mental health is particularly low (n=1116). Do you have any explanation for why this question was not answered by most of the participants? An explanation could be that employers might not have that information. They might only have information about sickness absence in general, but not about the reason for sickness absence.

On page 10, line 186 the authors write: “As pooled panel data were used, we did not employ specific strategies to address missing data in the analysis as any missing data points were inherently handled through the nature of the dataset.” Could you please explain, what is meant by “were inherently handled through the nature of the dataset”?

Response rate: As reported in the result section (page 11, line 202), the response rate for the four waves was between 17% (2020) and 15% (2021-2023). This low response rate together with the high degree of missing data among those who answered should be addressed as a limitation of the study in the section strengths and limitations.

It is however positive that the sample included a lot of small to medium sized workplaces (SMEs). In fact, around 80% of the participating organisations had less than 50 employees. Most studies about mental health at work are conducted in larger organisations, so it is clearly a strength that this study includes many SMEs that often have fewer resources to be active in mental health promotion. This strength could also be mentioned in the discussion part. Another strength is the mix of sectors, as there often is an overrepresentation of studies about mental health at work conducted in the health care sector. In this study, it is therefore good to see that sectors such as production and construction are represented. (See for example the study by Greiner et al. that calls for more research about mental health interventions in the construction sector. (Greiner, B. A., Leduc, C., O’Brien, C., Cresswell-Smith, J., Rugulies, R., Wahlbeck, K., ... & Aust, B. (2022). The effectiveness of organisational-level workplace mental health interventions on mental health and wellbeing in construction workers: A systematic review and recommended research agenda. Plos one, 17(11), e0277114.).

However, it would also be good to know how many organisations actually offered Line Manager training in mental health, that is, how was the question “LM training in mental health” (page 8, line 158) answered by the organisations? Could you report this and also shown it by size of organisation? This would be very valuable additional information for this topic: How widespread are LM training in mental health in these organisations (especially in in this sample representing sectors often not studied)? Are LM trainings just as common among SME businesses than they are among larger organisations? Please add this information.

Another issue that should be addressed is that at least some of the survey waves were conducted during the COVID-19 pandemic (see page 7, from line 131). The first wave (January to March 2020) was probably not affected, especially since the questions are about the past 12 months, so before the pandemic came to Europe. However, wave 2 and 3 conducted in 2021 and 2022 and thereby asking about 2020 and 2021 were conducted while most workplaces were affected by the pandemic in one way or the other. The authors probably have no specific knowledge about how this affected these workplaces, but it could be mentioned that the need for more awareness for employee mental health might have been even higher during these years (because for example more social isolation) and therefore it is good to see that the effects of this study were found despite of this. The topic is mentioned in the introduction but could also be mentioned again in the discussion with regard to the results.

I also noticed that the mean of recurrent long-term cases of sickness-absence due to mental health was 17% in 2020, but 49% in 2021, 45% in 2022 and 40% in 2023 (page 10, line 177). Could that maybe be seen as an increase in mental health problems severity among those that were suffering from mental health problems due to the pandemic? The proportion of sickness absence due to mental health only increased slightly during the four waves.

Minor comment: On page 11, line 202 the authors write “Response rate was calculated as the percentage of people who completed and returned the survey out of the total number of people invited to take part.” Was there something to return? Or should it be “answered” since the survey was done by telephone interview

We look forward to receiving your revised manuscript.

Kind regards,

Erum Shaikh

Academic Editor

PLOS ONE

Journal Requirements:

The data used here were originally collected as part of an Economic and Social Research Council funded project ‘Workplace mental-health and wellbeing practices, outcomes and productivity’ (Grant number: ES/W010216/1). This secondary analysis project ‘Mental health at work: a longitudinal exploration of line manager training provisions and impacts on productivity, individual and organisational outcomes’ was supported by the Economic and Social  Research Council [Productivity Institute: grant number: ES/V002740/1].

Reviewers' comments:

Reviewer's Responses to Questions

**Comments to the Author**

1. Is the manuscript technically sound, and do the data support the conclusions?

Reviewer #1: Yes

Reviewer #2: Yes

2. Has the statistical analysis been performed appropriately and rigorously? 

Reviewer #1: Yes

Reviewer #2: I Don't Know

3. Have the authors made all data underlying the findings in their manuscript fully available?

Reviewer #1: No

Reviewer #2: Yes

4. Is the manuscript presented in an intelligible fashion and written in standard English?

Reviewer #1: Yes

Reviewer #2: Yes

5. Review Comments to the Author

**Reviewer #1:** The paper "The Relationship between Line Manager Training in Mental Health and Organisational Outcomes" examines the effects of managerial training on organizational outcomes. While the introduction adequately establishes the research, incorporating a discussion on existing mental health training practices could improve its context. The literature review could be strengthened by including a variety of viewpoints and conflicting research according to the variables. The methodology section, centered around probit regression, could benefit from a discussion on its limitations and potential biases and it could be analyse using confirming statistical multivariate analysis methods. The results section is well-articulated and comprehensive, though adding visual data representations and concentrating data in tables can enhance understanding. The conclusions effectively recap the study’s findings; however, elaborating on their practical applications could enhance their utility in real-world settings; increasing references to previous studies are needed to confirm research assumptions. Finally, the references are comprehensive, yet updating them to include more of the latest studies would ensure the research's relevance and show the top industry standards.

Suggestions for Enhancement:

Introduction:

I'd like you to add insights on prevailing mental health training practices in the industry to anchor the research's significance better. Include a clear definition of your research questions, objectives, and hypothesis, perhaps in a table.

Literature Review:

Broaden the scope to include diverse research and a more comprehensive array of theoretical perspectives to enrich the narrative. Explain each of your research variables and define a table of authors where you include the top publications for each variable and dimension of your study.

Methodology:

Include more details on the selection of probit regression and address any inherent methodological weaknesses or biases. Explain why it was the best alternative for analyzing this data and if there's any other possible alternative method to validate your assumptions.

Results:

I'd like you to introduce graphical elements like charts or graphs to summarize and clarify key findings, compare your results with previous research, and determine if this study covers any gap in theory, methodology, or research methods.

Conclusions:

Detail the practical implications of the research findings, suggesting specific training or policy changes that could be derived from the study. Describe the stakeholder who are having the best use for your fingins. Describe how you achieve your results and goals.

References:

Update and verify that all references reflect the most current research, ensuring the study's alignment with the latest developments in the field.

**Reviewer #2: **This is an interesting study investigating the organizational level effects of line manager trainings for mental health. The authors found a way to assess these outcomes by analysing data from an existing dataset (panel survey data from firms in England) thereby adding important information about the effects of these trainings that go beyond an assessment by the participants.

The manuscript is well written and clearly structured and adds important information to this area of research.

However, I have some comments that I would like the authors to respond to before publication:

The authors are right in pointing out (page 5 line 90) that only few studies have examined the impact and influence of LM training on organisational-level outcomes (e.g., changes to productivity, turnover rates, and absenteeism). However, also the impact of LM training on employee outcomes is often not assessed. This also applies to the “Employee-level benefits” that the authors describe have been found (page 5 line 85). It does not look like employees have been asked about these benefits. Instead, the results describe managers’ own assessment of their gained competencies, their intentions to promote mental health at work and their increased confidence to support employees with mental health problems. I understand that this often is what can be measured in studies about manager training as asking employees would often result in a much larger study. However, since the authors address this topic I think it is important to mention that we actually know surprisingly little about the effects of LM trainings for employees, as most studies assess how the participants themselves (the managers) evaluate what they gained from having been trained. Results often show increased knowledge, confidence and good intentions, however, we most of the time do not know if this leads to actual implementing this in daily practice (for example more support for employees with mental health problems) and we know even less about if this leads to better outcomes for employees.

This lack of knowledge of manager training for employee outcome is discussed in for example the two papers below but can also be found in several reviews about manager training for employee mental health:

Rugulies, R., Aust, B., Greiner, B. A., Arensman, E., Kawakami, N., LaMontagne, A. D., & Madsen, I. E. (2023). Work-related causes of mental health conditions and interventions for their improvement in workplaces. The Lancet, 402(10410), 1368-1381.

Aust, B., Møller, J. L., Nordentoft, M., Frydendall, K. B., Bengtsen, E., Jensen, A. B., ... & Jaspers, S. Ø. (2023). How effective are organizational-level interventions in improving the psychosocial work environment, health, and retention of workers? A systematic overview of systematic reviews. Scandinavian Journal of Work, Environment & Health, 49(5), 315.

My next comment is about missing data. The authors report on page 11, line 200 that the sample consisted of 7139 participants after the four waves were merged. However, the numbers of answers for the different questions are much lower. In Table 3 the number of responses for the sickness absence related outcomes are between 1116 and 3566. In Table 4 the numbers for the responses for the organisational-level outcomes are between 3178 and 3189. Does that mean that only around half of the participants answered these questions? Please explain.

In addition, the number of answers for the questions about the proportion of sickness absence due to mental health is particularly low (n=1116). Do you have any explanation for why this question was not answered by most of the participants? An explanation could be that employers might not have that information. They might only have information about sickness absence in general, but not about the reason for sickness absence.

On page 10, line 186 the authors write: “As pooled panel data were used, we did not employ specific strategies to address missing data in the analysis as any missing data points were inherently handled through the nature of the dataset.” Could you please explain, what is meant by “were inherently handled through the nature of the dataset”?

Response rate: As reported in the result section (page 11, line 202), the response rate for the four waves was between 17% (2020) and 15% (2021-2023). This low response rate together with the high degree of missing data among those who answered should be addressed as a limitation of the study in the section strengths and limitations.

It is however positive that the sample included a lot of small to medium sized workplaces (SMEs). In fact, around 80% of the participating organisations had less than 50 employees. Most studies about mental health at work are conducted in larger organisations, so it is clearly a strength that this study includes many SMEs that often have fewer resources to be active in mental health promotion. This strength could also be mentioned in the discussion part. Another strength is the mix of sectors, as there often is an overrepresentation of studies about mental health at work conducted in the health care sector. In this study, it is therefore good to see that sectors such as production and construction are represented. (See for example the study by Greiner et al. that calls for more research about mental health interventions in the construction sector. (Greiner, B. A., Leduc, C., O’Brien, C., Cresswell-Smith, J., Rugulies, R., Wahlbeck, K., ... & Aust, B. (2022). The effectiveness of organisational-level workplace mental health interventions on mental health and wellbeing in construction workers: A systematic review and recommended research agenda. Plos one, 17(11), e0277114.).

However, it would also be good to know how many organisations actually offered Line Manager training in mental health, that is, how was the question “LM training in mental health” (page 8, line 158) answered by the organisations? Could you report this and also shown it by size of organisation? This would be very valuable additional information for this topic: How widespread are LM training in mental health in these organisations (especially in in this sample representing sectors often not studied)? Are LM trainings just as common among SME businesses than they are among larger organisations? Please add this information.

Another issue that should be addressed is that at least some of the survey waves were conducted during the COVID-19 pandemic (see page 7, from line 131). The first wave (January to March 2020) was probably not affected, especially since the questions are about the past 12 months, so before the pandemic came to Europe. However, wave 2 and 3 conducted in 2021 and 2022 and thereby asking about 2020 and 2021 were conducted while most workplaces were affected by the pandemic in one way or the other. The authors probably have no specific knowledge about how this affected these workplaces, but it could be mentioned that the need for more awareness for employee mental health might have been even higher during these years (because for example more social isolation) and therefore it is good to see that the effects of this study were found despite of this. The topic is mentioned in the introduction but could also be mentioned again in the discussion with regard to the results.

I also noticed that the mean of recurrent long-term cases of sickness-absence due to mental health was 17% in 2020, but 49% in 2021, 45% in 2022 and 40% in 2023 (page 10, line 177). Could that maybe be seen as an increase in mental health problems severity among those that were suffering from mental health problems due to the pandemic? The proportion of sickness absence due to mental health only increased slightly during the four waves.

Minor comment: On page 11, line 202 the authors write “Response rate was calculated as the percentage of people who completed and returned the survey out of the total number of people invited to take part.” Was there something to return? Or should it be “answered” since the survey was done by telephone interview

6. PLOS authors have the option to publish the peer review history of their article (what does this mean?). If published, this will include your full peer review and any attached files.

Reviewer #1: No

Reviewer #2: No

---

## [Author Response · Author response to Decision Letter 0]

16 May 2024

May, 2024.

Dear Editor,

Thank you for the opportunity to revise and resubmit our paper for PLOS ONE. We have responded to each comment below:

PONE-D-24-06886

The Relationship between Line Manager Training in Mental Health and Organisational Outcomes

Reviewer 1

The paper "The Relationship between Line Manager Training in Mental Health and Organisational Outcomes" examines the effects of managerial training on organizational outcomes. While the introduction adequately establishes the research, incorporating a discussion on existing mental health training practices could improve its context. The literature review could be strengthened by including a variety of viewpoints and conflicting research according to the variables. The methodology section, centered around probit regression, could benefit from a discussion on its limitations and potential biases and it could be analyse using confirming statistical multivariate analysis methods. The results section is well-articulated and comprehensive, though adding visual data representations and concentrating data in tables can enhance understanding. The conclusions effectively recap the study’s findings; however, elaborating on their practical applications could enhance their utility in real-world settings; increasing references to previous studies are needed to confirm research assumptions. Finally, the references are comprehensive, yet updating them to include more of the latest studies would ensure the research's relevance and show the top industry standards.

Thank you for these comments, we have responded to each suggestion below.

 

Introduction:

I'd like you to add insights on prevailing mental health training practices in the industry to anchor the research's significance better. 

We have integrated additional discussion regarding prevailing mental health training practises and industry. The particular and most prevalent form of training is mental health first aid. Although this is commonly observed in practice, its focus enrichment is notably different and unique from line manager training interventions - which we aim to draw out in this newly added text. 

Include a clear definition of your research questions, objectives, and hypothesis, perhaps in a table.

We have clearly defined our research questions and hypotheses at the end of the introduction (although we preferred to state these in the text rather than in a table, as the former is a more common approach).

Literature Review:

Broaden the scope to include diverse research and a more comprehensive array of theoretical perspectives to enrich the narrative. 

We have integrated in a discussion regarding the job demand resource model to provide an additional theoretical lens in relation to this study. In particular, our aim here is to highlight the conceptual understanding between working conditions, important mental health, and performance and productivity; and the theorised role played by mental health and well-being practises in facilitating these relationships. A paucity of research exists exploring the postulated link between employees’ well-being and organisational level outcomes within this model. Therefore, we believe this study provides an important contribution to this wider theoretical discussion and validation surrounding JDR. 

Explain each of your research variables and define a table of authors where you include the top publications for each variable and dimension of your study.

We have carefully considered this comment although the research team was unclear what the reviewer was asking for here.

Our Table 1 already includes each construct and its description. We understood the reviewer’s comment to be a request that we add the authors/citations for variables which are based on standardised measures. However, these items were developed by our project team and are not taken from standardised measures, and so there are no associated publications to add.

An alternative interpretation would be that the reviewer is asking for some kind of bibliometric analysis for each of the constructs measured, however this would be entirely new research (with its own aims and objectives) and is far beyond the scope of our study.

Methodology:

Include more details on the selection of probit regression and address any inherent methodological weaknesses or biases. Explain why it was the best alternative for analyzing this data and if there's any other possible alternative method to validate your assumptions.

Thank you for this, we have included some additional information in the Methods section. 

Results:

I'd like you to introduce graphical elements like charts or graphs to summarize and clarify key findings, compare your results with previous research, and determine if this study covers any gap in theory, methodology, or research methods.

Thank you for this. We believe this information is best presented in a Table. To compare our findings with previous research and determine the unique contributions of this research to the evidence-base, we have included a new Table in the results section.

Conclusions:

Detail the practical implications of the research findings, suggesting specific training or policy changes that could be derived from the study. Describe the stakeholder who are having the best use for your findings. Describe how you achieve your results and goals.

We have included the key stakeholders to whom are findings are relevant (i.e., business owners and managers, professional bodies, charities, and policymakers). Specific implications have been added including recommendations for organisations to establish policies and practices related to workplace mental health and line managers. We have signposted to current research in which relevant evidence-based line manager training has been developed.

References:

Update and verify that all references reflect the most current research, ensuring the study's alignment with the latest developments in the field.

This has been actioned.

Reviewer 2

This is an interesting study investigating the organizational level effects of line manager trainings for mental health. The authors found a way to assess these outcomes by analysing data from an existing dataset (panel survey data from firms in England) thereby adding important information about the effects of these trainings that go beyond an assessment by the participants. The manuscript is well written and clearly structured and adds important information to this area of research.

Thank you for this positive comment.

The authors are right in pointing out (page 5 line 90) that only few studies have examined the impact and influence of LM training on organisational-level outcomes (e.g., changes to productivity, turnover rates, and absenteeism). However, also the impact of LM training on employee outcomes is often not assessed. This also applies to the “Employee-level benefits” that the authors describe have been found (page 5 line 85). It does not look like employees have been asked about these benefits. Instead, the results describe managers’ own assessment of their gained competencies, their intentions to promote mental health at work and their increased confidence to support employees with mental health problems. I understand that this often is what can be measured in studies about manager training as asking employees would often result in a much larger study. However, since the authors address this topic, I think it is important to mention that we actually know surprisingly little about the effects of LM trainings for employees, as most studies assess how the participants themselves (the managers) evaluate what they gained from having been trained. Results often show increased knowledge, confidence and good intentions, however, we most of the time do not know if this leads to actual implementing this in daily practice (for example more support for employees with mental health problems) and we know even less about if this leads to better outcomes for employees.

We agree. Our study collected data from organisational representatives so we cannot comment on employee-level outcomes as this was not the focus of our study. 

However, it is an important gap in knowledge, and we have added text to the discussion:

“Fourth, most research on LM training in mental health generally explores outcomes for the manager (e.g., awareness, knowledge, confidence, skills), and our study provides novel data by focusing on outcomes at an organisational level (e.g., indices of business performance). However, few studies have explored the way in which LM training is implemented in practice, and whether (or not) this impacts on mental health at the employee-level. There is currently inconclusive evidence that leadership training impacts on employee outcomes, due to contradictory results (Aust et al, 2023). While employee outcomes were not the focus of our study, future research may seek to enhance the evidence-base relating to employee outcomes”. 

My next comment is about missing data. The authors report on page 11, line 200 that the sample consisted of 7139 participants after the four waves were merged. However, the numbers of answers for the different questions are much lower. In Table 3 the number of responses for the sickness absence related outcomes are between 1116 and 3566. In Table 4 the numbers for the responses for the organisational-level outcomes are between 3178 and 3189. Does that mean that only around half of the participants answered these questions? Please explain.

Thank you for this question. The questionnaire asked participants to indicate the proportions of the specific outcomes which occurred in their organisations (For e.g., sickness absence due to mental ill health). Not all the participating organisations indicated that they experienced this, hence, the corresponding numbers were smaller. For example, to give a response to the question ‘what is the proportion of sickness absence due to mental ill-health’, the participant first had to indicate that the organisation experienced sickness absence due to mental ill health. If they did not, this question would automatically be skipped – hence resulting in a smaller number of participants. For purposes of transparency, we reported these numbers to not mislead readers into thinking that the full sample size completed these questions. 

In addition, the number of answers for the questions about the proportion of sickness absence due to mental health is particularly low (n=1116). Do you have any explanation for why this question was not answered by most of the participants? An explanation could be that employers might not have that information. They might only have information about sickness absence in general, but not about the reason for sickness absence.

The reason for the smaller sample size here is the same as the reason given above. Either the participant reported that the organisation did not have sickness absence due to mental ill health or as you suggested, it is possible that they did not have the information and hence the question was skipped. 

On page 10, line 186 the authors write: “As pooled panel data were used, we did not employ specific strategies to address missing data in the analysis as any missing data points were inherently handled through the nature of the dataset.” Could you please explain, what is meant by “were inherently handled through the nature of the dataset”?

Pooled panel data typically combines both time series data and cross sectional data into a single dataset. As mentioned in the Methods section, strategically pooling this data together granted us greater statistical power which meant that missing data from a few points would not have a significant impact on the overall variance and mean estimates, hence eliminating the need for missing data techniques. 

Response rate: As reported in the result section (page 11, line 202), the response rate for the four waves was between 17% (2020) and 15% (2021-2023). This low response rate together with the high degree of missing data among those who answered should be addressed as a limitation of the study in the section strengths and limitations.

We have now included this as a limitation of the study. 

It is however positive that the sample included a lot of small to medium sized workplaces (SMEs). In fact, around 80% of the participating organisations had less than 50 employees. Most studies about mental health at work are conducted in larger organisations, so it is clearly a strength that this study includes many SMEs that often have fewer resources to be active in mental health promotion. This strength could also be mentioned in the discussion part. Another strength is the mix of sectors, as there often is an overrepresentation of studies about mental health at work conducted in the health care sector. In this study, it is therefore good to see that sectors such as production and construction are represented. (See for example the study by Greiner et al. that calls for more research about mental health interventions in the construction sector. (Greiner, B. A., Leduc, C., O’Brien, C., Cresswell-Smith, J., Rugulies, R., Wahlbeck, K., ... & Aust, B. (2022). The effectiveness of organisational-level workplace mental health interventions on mental health and wellbeing in construction workers: A systematic review and recommended research agenda. Plos one, 17(11), e0277114.).

Thank you for raising this. We have added text to the study strengths section and included the suggested reference:

“A significant strength of the study is its large sample size and diversity of organisations included. We are therefore able to report findings across sectors and sizes of organisations, including sectors in which mental ill-health is prevalent, but research evidence relating to workplace mental health interventions is limited (e.g., construction: [Greiner et al, 2022]). Another strength of the study is that around 80% of participating organisations had fewer than 50 employees; our study therefore includes many SMEs that are under-researched and often have fewer resources to be active in mental health promotion”.

However, it would also be good to know how many organisations actually offered Line Manager training in mental health, that is, how was the question “LM training in mental health” (page 8, line 158) answered by the organisations? Could you report this and also shown it by size of organisation? This would be very valuable additional information for this topic: How widespread are LM training in mental health in these organisations (especially in in this sample representing sectors often not studied)? Are LM trainings just as common among SME businesses than they are among larger organisations? Please add this information.

An analysis on the same dataset has recently been published providing this information. We have now cited this article and referred to it in the introduction (showing the % increase from 2020-2023) and methods:

Blake H, Hassard J, Dulal-Arthur T, Wishart M, Roper S, Bourke J, Belt V, Bartle C, Pahl N, Leka S, Thomson L. Typology of employers offering line manager training for mental health. Occupational Medicine, 2024;74(3):242–250, https://doi.org/10.1093/occmed/kqae025

Another issue that should be addressed is that at least some of the survey waves were conducted during the COVID-19 pandemic (see page 7, from line 131). The first wave (January to March 2020) was probably not affected, especially since the questions are about the past 12 months, so before the pandemic came to Europe. However, wave 2 and 3 conducted in 2021 and 2022 and thereby asking about 2020 and 2021 were conducted while most workplaces were affected by the pandemic in one way or the other. The authors probably have no specific knowledge about how this affected these workplaces, but it could be mentioned that the need for more awareness for employee mental health might have been even higher during these years (because for example more social isolation) and therefore it is good to see that the effects of this study were found despite of this. The topic is mentioned in the introduction but could also be mentioned again in the discussion with reg

---

## [Editor Report · Decision Letter 1]

11 Jun 2024

The Relationship between Line Manager Training in Mental Health and Organisational Outcomes

PONE-D-24-06886R1

Dear Dr. Blake,

We’re pleased to inform you that your manuscript has been judged scientifically suitable for publication and will be formally accepted for publication once it meets all outstanding technical requirements.

Kind regards,

Erum Shaikh

Academic Editor

PLOS ONE
---

## [Editor Report · Acceptance letter]

24 Jun 2024

PONE-D-24-06886R1 

PLOS ONE

Dear Dr. Blake, 

I'm pleased to inform you that your manuscript has been deemed suitable for publication in PLOS ONE. Congratulations! Your manuscript is now being handed over to our production team.

Kind regards, 

on behalf of

Dr. Erum Shaikh 

Academic Editor

PLOS ONE